



# Technical note: What does the Standardized Streamflow Index actually reflect? Insights and implications for hydrological drought analysis

Fabián Lema[1], Pablo A. Mendoza[1,2], Nicolás A. Vásquez[1], Naoki Mizukami[3], Mauricio Zambrano-Bigiarini[4,5] and Ximena Vargas[1]

[1]Department of Civil Engineering, Universidad de Chile, Santiago, Chile
[2]Advanced Mining Technology Center, Universidad de Chile, Santiago, Chile
[3]Research Applications Laboratory, National Center for Atmospheric Research, Boulder, Colorado, USA
[4]Department of Civil Engineering, Universidad de La Frontera, Temuco, Chile
[5]Center for Climate and Resilience Research, Universidad de Chile, Santiago, Chile

*Correspondence to:* Pablo A. Mendoza (pamendoz@uchile.cl)

**Abstract.** Hydrological drought is one of the main hydroclimatic hazards worldwide, affecting water availability, ecosystems and socioeconomic activities. This phenomenon is commonly characterized by the Standardized Streamflow Index (SSI), which is widely used because of its straightforward formulation and calculation. Nevertheless, there is limited understanding of what the SSI actually reveals about how climate anomalies propagate through the terrestrial water cycle. To find possible explanations, we implemented the SUMMA hydrological model coupled with the mizuRoute routing model in six hydroclimatically different case study basins located on the western slopes of the extratropical Andes, and examined correlations between the SSI (computed from the models for 1, 3 and 6-month time scales) and potential explanatory variables – including precipitation and simulated catchment-scale storages – aggregated at different time scales. Additionally, we analyzed the impacts of adopting commonly used time scales on propagation analyses of specific drought events – from meteorological to soil moisture and hydrological drought – with focus on their duration and intensity. The results reveal that the choice of time scale for the SSI has larger effects on correlations with explanatory variables in rainfall-dominated regimes compared to snowmelt-driven basins, especially when simulated fluxes and storages are aggregated to time scales longer than 9 months. In all the basins analyzed, the strongest relationships (Spearman rank correlation values over 0.7) were obtained when using 6-month aggregations to compute the SSI and 9-12 months to compute the explanatory variables, excepting aquifer storage in snowmelt-driven basins. Finally, the results show that the trajectories of drought propagation obtained with the Standardized Precipitation Index (SPI), the Standardized Soil Moisture Index (SSMI) and the SSI may change drastically with the selection of time scale. Overall, this study highlights the need for caution when selecting standardized drought indices and associated time scales, since their choice impacts event characterizations, monitoring and propagation analyses.



# 1 Introduction

Droughts are natural hazards that can cover vast areas over a period of months to several years (Samaniego et al., 2013; Brunner & Tallaksen, 2019), with large effects on environmental systems (Vicente-Serrano et al., 2020) and socioeconomic activities

(Wilhite & Pulwarty, 2017). These events are primarily triggered by precipitation deficits (McKee et al., 1993), which may be associated to internal climate variability modes – such as El Niño Southern Oscillation (Okumura et al., 2017; Steiger et al., 2021) – and exacerbated by land-atmosphere interactions (Schumacher et al., 2022). Given the warming trends projected for the next decades (e.g., Brunner et al., 2020; Tokarska et al., 2020) and the contribution of higher temperature to drying (Trenberth et al., 2014), anthropogenic climate change is also expected to affect drought characteristics, increasing their

frequency, severity, and duration in many regions of the world (e.g., Cook et al., 2014; Boisier et al., 2016; Pokhrel et al., 2021).

Despite the drought concept refers to the notion of below-average water fluxes and/or storages (Tallaksen & Van Lanen, 2004; Van Loon, 2015), there are several drought definitions and classifications, being meteorological (below-normal precipitation), soil moisture (also referred to as agricultural drought; e.g., Thober et al., 2015; Cook et al., 2018), hydrological (surface and

groundwater level deficits), and socioeconomic the most commonly used types (Wilhite & Glantz, 1985). Among these, hydrological droughts – associated with abnormally low levels in surface water bodies, groundwater and/or streamflow in rivers (Van Loon, 2015) – are especially relevant due to their direct impacts on natural ecosystems and human society. Hence, understanding how climate anomalies propagate through the terrestrial water cycle to trigger hydrological droughts of different characteristics (e.g., duration, severity) is an outstanding challenge for the scientific community, and a crucial task for water

resources planning and management (Zhang et al., 2022).

Hydrological droughts are typically quantified through indices derived from observed or modeled time series of streamflow (e.g., Zhu et al., 2016; Stahl et al., 2020), runoff (Shukla and Wood, 2008), and groundwater levels (e.g., Bachmair et al., 2015). Among the existing indices, the Standardized Streamflow Index (SSI; Vicente-Serrano et al., 2012) has become increasingly popular because of its straightforward formulation, calculation, and interpretability for the characterization of

discharge anomalies. In fact, a search in the Clarivate Analytics Web of Science platform using the keywords "standardized streamflow index" (or "standardised streamflow index") and "drought" revealed 163 journal articles at the time of this writing. Such body of work spans various areas, including drought monitoring (Núñez et al., 2014; Nkiaka et al., 2017) and forecasting (e.g., Sutanto & Van Lanen, 2021; Sutanto & Van Lanen, 2022; Hameed et al., 2023), as well as drought propagation under historically observed (e.g., Barker et al., 2016; Bhardwaj et al., 2020) and projected (Wan et al., 2018; Adeyeri et al., 2023)

climatic conditions.

The applicability of the SSI is challenged by its sensitivity to the quantity and quality of the data (Wu et al., 2018) and the calculation method, which entails the choice of a reference period for standardization, the selection of probability distribution (e.g., Laimighofer and Laaha, 2022; Teutschbein et al., 2022), the parameter estimation approach (e.g., Tijdeman et al., 2020) and, in particular, the time scale or accumulation (e.g., Barker et al., 2016; Baez-Villanueva et al., 2024). The latter refers to





the backward-looking period (commonly a number of months) over which streamflow values are averaged before computing
the index. Most drought propagation analyses seek possible relationships between meteorological drought indices (e.g., McKee
et al., 1993; Vicente-Serrano et al., 2010) computed for various time scales and the SSI for some time scale, being one month
(SSI-1) the most common choice (e.g., Huang et al., 2017; Peña-Gallardo et al., 2019; Stahl et al., 2020; Wang et al., 2020;
Wu et al., 2022; Zhang et al., 2022; Odongo et al., 2023; Baez-Villanueva et al., 2024). Such decision commonly relies on the

assumption that streamflow already includes hydro-meteorological processes of the previous months (e.g., Stahl et al., 2020;
Tijdeman et al., 2020; Sutanto and Van Lanen, 2021), enabling direct comparisons with them (e.g., Baez-Villanueva et al.,
2024). Because the SSI-1 may be susceptible to short-term fluctuations, other authors have preferred smoothed (e.g., 3-month
averages) time series of SSI-1 (e.g., Bhardwaj et al., 2020), 3-month (e.g., Núñez et al., 2014; Wu et al., 2017; Rivera et al.,
2021; Adeyeri et al., 2023; Yun et al., 2023), 6-month (e.g., Seibert et al., 2017; Oertel et al., 2020), or even longer (e.g.,

Teutschbein et al., 2022; Fowé et al., 2023) time scales.

Nowadays, there is no consensus regarding the most appropriate time scale for both SSI and possible explanatory variables
(e.g., precipitation and catchment-scale simulated storages), which may stem from the limited understanding of what the SSI
truly reveals about the underlying physical mechanisms driving hydrological droughts. For example, Buitink et al. (2021)
examined five components of the water cycle – precipitation, soil moisture, vegetation greenness, groundwater and surface

water – in the Dutch province of Gelderland, finding that percentile-based thresholds that are commonly used for hydrological
drought detection mask out more frequent drought conditions that other variables in the system may be experiencing.

To tackle this issue, process-based hydrological modeling arises as a useful approach (Peters-Lidard et al., 2021), and the
literature is rich in studies using models with varying degrees of complexity to examine the propagation from meteorological
to soil moisture or hydrological droughts (e.g., Andreadis et al., 2005; Sheffield & Wood, 2007; Van Loon & Van Lanen,

2012; Samaniego et al., 2013; Van Loon et al., 2014; Zink et al., 2016; Apurv et al., 2017; Bhardwaj et al., 2020; Lee et al.,
2022; Rakovec et al., 2022). This paper contributes to this field by combining observed data and a state-of-the-art physics-
based modeling framework to analyze fluctuations in the widely used SSI across hydrological regimes. Here, we depart from
previous hydrological drought assessments by first conducting exploratory correlation analyses between modeled catchment-
scale water storages and the SSI, to subsequently inform the choice of time scales for the calculation of standardized indices

(e.g., Samaniego et al., 2013) to perform drought propagation analyses. Specifically, we address the following research
questions:

    1. What are the effects of different time scales on the number and duration of hydrological droughts?

    2. How does the SSI relate to catchment-scale water storages and fluxes across different hydrological regimes?

    3. How do different time scales affect the propagation of historically observed meteorological droughts towards soil

95       moisture and hydrological droughts?

To seek for answers, we configure the Structure for Unifying Multiple Modeling Alternatives (SUMMA; Clark et al., 2015a,
2015b) hydrological model and the vector-based routing model mizuRoute (Mizukami et al., 2016, 2021) in six basins located
along the western slopes of the extratropical Chilean Andes. Catchment-scale precipitation and model simulations are





temporally aggregated to monthly time steps to compute snow water equivalent (SWE), soil moisture, aquifer storage, total
storage (i.e., the sum of SWE, soil moisture, aquifer storage, and canopy storage) and the SSI for different time scales. We use
these time series to explore the physical processes explaining variations in the SSI during the period April/1983-March/2020,
as well as the drought event of 1998/99 and the recent central Chile megadrought (Garreaud et al., 2017; Garreaud et al., 2019).
Finally, we examine the implications of time scale selection on the portrayal of drought propagation across the duration-
intensity space, using standardized indices during historically observed events. We stress that it is not our intention to select
or establish the most suitable time scale to be applied in each particular case; instead, we seek to improve the current
understanding of the information content of the SSI across different hydrological regimes and raise awareness on the impact
that the subjective choice of the time scale and the analysis periods may have on the interpretation and application of the SSI
for drought monitoring and propagation analyses.

## 2 Study Area and data

### 2.1 Case study basins

We conduct our analyses in six Chilean basins located on the western slopes of the extratropical Andes Cordillera (Figure 1):
(i) Cochiguaz River at El Peñón, (ii) Choapa River at Cuncumén, (iii) Claro River at El Valle, (iv) Palos River at Colorado,
(v) Ñuble River at La Punilla, and (vi) Cautín River at Rari-Ruca. Hereafter, to refer to each basin using the name of the river.
The catchment boundaries and the identification number (ID) are obtained from the CAMELS-CL database (Alvarez-Garreton
et al., 2018). All the basins receive most of the precipitation during the Fall (MAM) and Winter (JJA) seasons (Figure 1).
Additionally, the basins span a wide range of physiographic characteristics and climatic conditions, with annual precipitation
amounts ranging from 260 to 2900 mm/year, mean annual temperatures between 9 to 16 °C, annual runoff spanning 114-2090
mm/year, and aridity indices between 0.4 to 3 (Table 1). Such climatic diversity translates into different hydrological regimes:
the Cochiguaz and Choapa River basins are snowmelt-driven, Palos and Ñuble have a mixed regime, while Claro and Cautín
are mostly rainfall-driven.

### 2.2 Datasets

Meteorological daily data are obtained from the CR2MET v.2.0 observational product (DGA, 2017; Boisier et al., 2018),
which provides precipitation and extreme temperature estimates for the period 1979-2020 at a 0.05° x 0.05° horizontal
resolution. CR2MET precipitation estimations are obtained through multiple linear regression models that consider
physiographic attributes and large-scale climate variables from the fifth generation of the European Reanalysis (ERA5;
Hersbach et al., 2020) as predictors, and observed daily precipitation from gauge stations as predictands. For extreme daily
temperatures, land surface temperature from the Moderate Resolution Imaging Spectroradiometer (MODIS) is included as a
potential explanatory variable. Daily precipitation and temperature time series are disaggregated into hourly time steps using
the sub-daily distribution provided by ERA5-Land (Muñoz-Sabater et al., 2021). Wind, incoming shortwave radiation,





atmospheric pressure, and relative humidity are obtained from ERA5-Land, whereas incoming longwave radiation is computed using the formulation proposed by Iziomon et al. (2003). Land cover data and vegetation types for the study area are also obtained from MODIS. Daily streamflow records are collected by the Chilean Water Directorate (DGA), and were retrieved from the website of the Climate and Resilience Research Centre (CR2,  https://www.cr2.cl/datos-de-caudales/).

## 3 Approach

Our approach considers the configuration of the SUMMA hydrological model and the mizuRoute routing model (Figure 2a); the calibration and evaluation of the SUMMA model parameters (Figure 2b); the computation of standardized drought indices (SDIs) for precipitation, simulated soil moisture and simulated streamflow (Figure 2c), as well as the analysis of their interplay with other simulated hydrological variables (Figure 2d). Finally, we examine how time scales typically adopted for the calculation of standardized indices affect the portrayal of historically observed drought events (Figure 2e). A key aspect of our

methodology is the identification of hydrological variables and time scales driving fluctuations in the SSI, obtaining all the data from a calibrated, state-of-the-art process-based hydrological model. This approach departs from previous efforts searching for statistical relationships between the SSI – computed with streamflow observations – and standardized indices such as the SPI (e.g., Barker et al., 2016; Huang et al., 2017; Wu et al., 2022), the SPEI (e.g., Peña-Gallardo et al., 2019; Wang et al., 2020; Bevacqua et al., 2021), or other indices and state variables derived from reanalysis datasets that do not necessarily

correspond to observed streamflow anomalies (e.g., Hoffmann et al., 2020; Baez-Villanueva et al., 2024).

### 3.1 Hydrological modeling

We use the SUMMA hydrologic modeling system, which offers different implementations for a wide range of modeling decisions. Specifically, SUMMA has several options for model configuration, process representations, and flux parameterizations for mass and energy balance equations. Here, we used Jarvis (1976) function for simulating stomatal

resistance, one of the main physiological factors controlling transpiration, similar to the Noah-MP land surface model (Niu et al., 2011). We also considered a logarithmic wind profile below the vegetation canopy – described in Mahat et al. (2013) –, and implemented the Raupach (1994) parameterization for vegetation roughness length and displacement height. We use Beer's law (Mahat and Tarboton, 2012) – as implemented in the Variable Infiltration Capacity (VIC) model (Liang et al., 1994) – to represent the radiation transmission through vegetation. For the vertical redistribution of water along the soil

column, we considered the mixed form of the Richards equation (Celia et al., 1990), a vertically constant hydraulic conductivity and a lumped aquifer model. For snow, we consider a constant albedo decay rate, and the thermal conductivity was parameterized using the Jordan (1991) approach.

In this study, each basin is spatially discretized into grid cells that are delineated to match the meteorological forcing data resolution (0.05° x 0.05°). Each grid cell has specific physiographic characteristics (e.g., slope, elevation, layer thickness, vegetation, and soil type), a maximum of five snow layers, and three soil layers with different thicknesses – top: 0.5 m, middle:


2 m, bottom: 2.5 m –. Further, each grid cell incorporates an unconfined aquifer at the bottom of the soil column, which contributes to baseflow generation (Figure 2a). We stress that no lateral water fluxes are allowed between grid cells.

We use the vector-based routing model mizuRoute (Mizukami et al., 2016, 2021) to convert the instantaneous runoff obtained with the SUMMA model at each grid cell into streamflow at the basin outlet. The application of mizuRoute requires delineating

a digital river network, with individual subcatchments contributing runoff to each river reach. First, the model converts the total runoff from each grid cell into subcatchment-scale runoff using area-weighted averages. Then, the model performs a hillslope routing to delay instantaneous total runoff from the subcatchment to the corresponding outlet using a gamma-distribution-based unit hydrograph, and then routes the delayed runoff for each river reach in the order defined by the river network topology. Full descriptions of the hillslope routing, general routing procedures, and routing schemes are provided by

Mizukami et al. (2016). Here, we use the Diffusive Wave routing scheme described and implemented by Cortés-Salazar et al. (2023).

## 3.2 Model calibration and evaluation

We calibrated 14 parameters (Table S1 in the Supplement) of the SUMMA model using the Dynamically Dimension Search algorithm (DDS; Tolson and Shoemaker, 2007) implemented in the OSTRICH software (Matott, 2017), with 2000 iterations

in each trial. We maximize the objective function (OF) proposed by Garcia et al (2017), which is focused on low-flows:

$$OF = 0.5 \cdot KGE(Q) + 0.5 \cdot KGE(1/Q) \tag{3.1}$$

where KGE is the Kling-Gupta efficiency (Gupta et al., 2009). The observed streamflow data is split into a calibration period (April/2010 – March/2017) and two non-consecutive evaluation periods (April/2006 – March/2010 and April/2017 – March/2020). For model evaluation, we use the OF, KGE, the Nash-Sutcliffe efficiency (NSE; Nash and Sutcliffe, 1970), the

coefficient of determination ($R^2$), and the root mean square error (RMSE, Figure 2b).

## 3.3 Drought analysis

### 3.3.1 Drought indices

For meteorological drought characterization, we use the well-known Standardized Precipitation Index (SPI; McKee et al., 1993). The SPI compares the cumulative precipitation for a specific temporal scale with its long-term (usually 30 years or

more) distribution at a given location. The SPI calculation involves (i) selecting a probability density function (PDF) and its parameters to obtain the reference long-term distribution for cumulative precipitation; (ii) obtaining the cumulative distribution function (CDF) from the fitted distribution; and (iii) transforming the CDF into a standardized normal distribution (i.e. with mean equal to zero and standard deviations of one), using an equi-percentile inverse transformation to derive the SPI values. Here, we use the parametric Gamma distribution (e.g., McKee et al., 1993; Stagge et al., 2015) and the probability-weighted

moments method (Hosking, 1986) to estimate its parameters in SPI calculations. We also use the Standardized Precipitation





and Evapotranspiration Index (SPEI; Vicente-Serrano et al., 2010), which requires monthly precipitation and temperature data and involves a mass balance given by the difference between precipitation and potential evapotranspiration (*PET*) estimated with the Thornthwaite (1948) equation.

For soil moisture drought analysis, we use the Standardized Soil Moisture Index (SSMI; Carrão et al., 2013), which quantifies
deficits in the soil water content in the root zone relative to its seasonal climatology at a specific location. The SSMI uses an empirical distribution based on monthly soil moisture series. Since the SUMMA model provides other storages besides soil moisture, we also propose and use a modified version – the Standardized Water Storage Index (SWSI) – to assess total water storage (i.e., the sum of SWE, canopy storage, soil moisture, and aquifer storage). Finally, we use the Standardized Streamflow Index (SSI; Vicente-Serrano et al., 2012) for hydrological drought characterization. Here, we use the generalized logistic
distribution to compute the SSI, following recommendations from past studies (e.g., Vicente-Serrano et al., 2012; Tijdeman et al., 2020).

To evaluate how the subjective choice of time scales may affect the characterization of different types of droughts and inter-relationships, we compute SDI-n with n = 1, 3, 6, 9, 12, 18, and 24 months (Figure 2c), excepting the SSI. While a monthly scale has been the most common choice for this index (SSI-1; e.g., Stahl et al., 2020; Baez-Villanueva et al., 2024), we consider
longer time scales that have been adopted in previous studies under different assumptions and considerations (e.g., Oertel et al., 2020; Tijdeman et al., 2020; Adeyeri et al., 2023; Fowé et al., 2023). We further examine how different drought detection criteria may alter the frequency and intensity of hydrological droughts events during a historical period. To this end, we apply a fixed threshold criterion (Van Loon, 2015) – set here as -1 – in two different ways: (i) a drought event starts when SDI-n drops below -1 and ends when it reaches or exceeds -1; (ii) a drought event begins when SDI-n remains below -1 for at least
three consecutive months and concludes when it reaches or exceeds -1.

### 3.3.2 Correlation analysis

To understand temporal fluctuations in the SSI, we compute the Spearman's rank correlation coefficient between SSI-n with n = 1, 3, and 6 months, which are the most commonly used temporal scales in drought propagation analyses (e.g., Núñez et al., 2014; Oertel et al., 2020), and the main catchment-scale water fluxes and storages as explanatory variables (Figure 2d),
including precipitation, SWE, soil moisture, aquifer storage and the total water storage in the basin (i.e., the sum of SWE, canopy storage, soil moisture and aquifer storage). To assess what time scale of the hydrological variables are important for drought occurrence, we use temporal averages or accumulations over the preceding months of 1, 3, 6, 9. 12, 18, and 24 (including the target month). In this analysis, we assume that the factors not simulated by the hydrological and routing models (e.g., land cover change, water abstractions, glaciers) have negligible influence on hydrological drought occurrence in the
selected basins.

The correlation analyses were conducted independently at each study basin over different temporal windows that include exceptionally dry water years. The goal here is to identify the strongest relationships between the SSI and explanatory





variables, the associated temporal scales, and whether these vary substantially with hydrological regimes and/or drought events.

### 3.3.3 Drought propagation analysis

Using the time scales that maximize correlations identified in section 3.3.2, we compute the SPI, SSMI, and SSI indices to characterize the transition from meteorological to hydrological droughts, passing through soil moisture drought (SPI → SSMI → SSI; Figure 2e). Here, we analyze the duration (in months) and the intensity, quantified as the temporally-averaged index value during its respective drought duration. We also compare the drought propagation portrayals derived from the time scales identified here, against other criteria adopted in recent studies (Table 2). These include propagation analyses using one (Wan et al., 2018) and three-month (e.g., Gautam et al., 2024) time scales for SPI, SSMI, and SSI calculations, as well as varying time scales for these indices depending on the hydrological regime of the target basin (e.g., Baez-Villanueva et al., 2024).

## 4 Results

### 4.1 Hydrological model performance

Figure 3 displays hydrological model calibration and evaluation results for the six study basins, showing an overall good agreement between observed and simulated streamflows. The value of the objective function (Eq 3.1) during the evaluation period is higher than 0.73 in all basins (Figure 3a). The minimum KGE during the calibration period is 0.74 (Choapa), whereas the highest KGE values are 0.83 (Palos) and 0.82 (Cautín). Negative biases (i.e., underestimation of runoff volumes) are obtained for Cochiguaz (-15.4%) and Ñuble (-5.8%), while small (< 8%) positive biases are obtained in the remaining basins. In general, the observed daily flow duration curves are well simulated by the SUMMA model in all catchments (Figure 3b), including its midsegment slope (20% - 70% flow exceedance probabilities); nevertheless, there is an overestimation of low flow volumes with exceedance probabilities larger than 90% in the Choapa and Claro catchments (< 2 m$^3$/s). The streamflow seasonality is well reproduced by the SUMMA model in all basins (Figure 3c), though there is an overestimation (< 10%) of mean monthly flows during September-November (i.e., when snowmelt occurs) at the Choapa and Claro River basins, and during March-October (i.e., when rainfall events occur) at the Ñuble and Cautin River basins.

### 4.2 Effects of time scale on drought characteristics

Figure 4 illustrates the time series for different simulated hydrological variables, as well as the SPI and SSI indices computed at different time scales for the Choapa and Cautín River basins. We focus on a three-year period (1998-2000) that includes the year 1998, a remarkably dry year spanning a 6-month period (July-December) with abnormally low precipitation amounts (Kreibich et al., 2022). Such precipitation deficit had a noticeable impact on snow accumulation, especially in the Choapa River basin (snowmelt-driven), and affected other variables to a lesser degree, including soil moisture (agricultural drought)




and aquifer storage, whose levels were even lower than those recorded in subsequent years. Ultimately, the meteorological drought translated into lower streamflow values over the course of 1998 and even 1999.

Figures 4b and 4g show the impacts of time scale selection on the SSI and the SPI, with substantial differences between 1-
month indices and aggregations larger than 12 months (18 and 24 months). This is especially noticeable in the SSI time series of the Choapa River basin, where a similar behavior over time is observed for SSI-1, SSI-3, SSI-6, and SSI-9, with index values smaller than –1 between October/1998 and September/1999. Nevertheless, the onset of hydrological drought is detected in May/1999 (end of 1999) if an 18-month (24-month) time scale is used to compute the SSI. Notably, Figure 4g shows that even a 1-month streamflow aggregation period in SSI calculations can distort the actual variability of streamflow considerably.

The choice of time scales used to compute the SSI can also affect the estimated frequency and duration of hydrological drought events. This is illustrated in Figure 5, which compares the number of hydrological drought events detected with SSI-1, SSI-3, and SSI-6, as well as the probabilistic distribution of their duration over the entire simulation period (April/1983 – March/2020). The results are displayed according to two criteria: (1) no constrains regarding the duration of droughts – i.e., it is possible to detect one-month events ("free") –, and (2) establishing a minimum duration of three months ("constrained") –
i.e., SSI-n < -1 during at least three consecutive months. Figure 5a shows substantial differences in the number of events depending on the criteria and time scale used, with the only exception being the Cochiguaz River basin. In general, the number of events detected with the free criterion decreases for longer temporal aggregations, as opposed to the constrained criterion, for which such number tends to remain constant or even increase (see, for example, the Choapa River basin). The largest discrepancies are found in the rainfall-dominated catchments; for example, in the Cautín River basin 28 and 13 events were
detected with the SSI-1 and SSI-6, respectively, using the free criterion. Figures 5b and 5c display the empirical probability density functions of drought durations obtained with the free and constrained criteria, for all basins and time scales, and Table 3 includes the average durations considering all the events during the analysis period. As for the frequency, we found no changes in the Cochiguaz River basin; however, the choice of time scale has considerable effects on drought durations in rainfall-driven catchments – especially with the free criterion –, with a transition from positively skewed probability density
functions with averages between 1-3 months when using SSI-1, to more homogeneous distributions – centered around 8 months – when using SSI-6.

### 4.3 Correlation between the SSI and hydrological variables

Figure 6 illustrates how the choice of time scale used for the SSI affects the Spearman rank correlation between this index and the main hydrological variables (precipitation and catchment storages). One can note that the differences are minimal between
the SSI-1, SSI-3, and SSI-6 for the two snowmelt-driven basins (i.e., Cochiguaz and Choapa). Further, the shape of the curves is similar in most cases, achieving the highest correlations with precipitation and SWE on a 12-month scale, and the highest correlations with soil moisture and total storage using time scales between 6 and 12 months. Notably, the strength of the relationship between the SSI and aquifer storage varies depending on the hydrological regime: in snowmelt driven basins, the





correlations are higher for time scales of 3-6 months of aquifer storage, whereas correlation is maximized with 9-12-month
time scales in rainfall-dominated catchments.

In most cases, the highest (lowest) correlations are obtained using SSI-6 (SSI-1), although there are some exceptions for time scales shorter than 9 months at the Palos and Ñuble River basins (mixed regime), where higher correlations are achieved when using SSI-1. The impacts of the time scale on correlation results are considerably larger in basins with mixed or rainfall-dominated regimes, where there is larger dispersion in the correlation achieved by the indices, reaching differences up to 0.5 in the Ñuble and Cautín River basins for a 12-month scale. Similarly, a progressive increase in the dispersion of correlations is observed when evaluating indices at larger time scales (> 9 months) for all storages in mixed and rainfall-driven catchments. Overall, the results in Figure 6 suggest that – if the aim is to investigate the relationship between the main hydrological variables and fluctuations in the SSI – the choice of the time scale used to compute this index becomes less relevant in snowmelt driven basins with large baseflow contributions, compared to rainfall-dominated catchments.

We now explore potential effects of hydrological regimes and the choice of analysis period on the relationship between SSI-n, precipitation and the main water storages at the catchment scale. Figure 7 displays Spearman rank correlations between the SSI-6 and hydrological variables aggregated at different time scales, for the six study basins and three periods: the 1998/1999 drought event, the central Chile megadrought (2010-2019), and April/1983 - March/2020 (the results for SSI-1 and SSI-3 are presented in Figures S1 and S2 of the Supplement). The examination of different storages reveals that, in general, higher Spearman rank correlations are obtained in arid and snowmelt-driven basins compared to humid and rainfall-driven basins, regardless of the time scale analyzed. In other words, there are stronger relationships with SSI-6 in the northern regions (aridity index > 2 and mean annual P < 400 mm/yr), which gradually become weaker towards the south (aridity index < 0.5 and mean annual P > 2000 mm/yr), following the central Chile's hydroclimatic gradient. Such pattern is more evident when all catchment storages are aggregated (last row in Figure 7) and to a smaller degree in individual storages (SWE, soil moisture, and aquifer storage).

Figure 7 also shows that the magnitude of correlations between hydrological variables and SSI-6 varies with the analysis period, especially during exceptionally dry periods. For example, the relationships between SSI-6 and precipitation in rainfall-dominated and mixed regime catchments (Claro, Palos, Ñuble and Cautín) are stronger during the 1998/99 drought, with Spearman rank correlations near 1 for a 9-month scale, whereas the remaining periods yield correlations that do not exceed 0.7 at the same temporal scale. Considerable differences are also obtained for SWE, with high correlations (>0.7) in all basins for the 9 and 12-month time scales during the 1998/99 event, and lower correlations over larger time windows (the whole period). The selection of analysis period also yields clear differences in the correlation between soil moisture and aquifer storage. Indeed, higher correlations are obtained for these variables when using time scales shorter than 9 months, especially in Choapa and Palos, where the snowmelt contribution to runoff is substantial.



## 4.4 Effects of temporal scale on drought propagation

To what extent can the choice of the temporal scale affect the portrayal of drought propagation across different hydrological regimes? To find answers, we examine the transition of meteorological towards soil moisture and hydrological droughts in the duration–intensity space using four different criteria, with a focus on two events – the 1998/99 and 2012-2016 droughts (a subperiod of the Chilean megadrought) – that simultaneously affected the Choapa (snowmelt driven), Palos (mixed regime) and Cautín (rainfall driven) River basins (Figure 8). The same analysis was also performed for the Cochiguaz, Claro, and Ñuble River basins, which are included in Figure S3 of the Supplement. The results show that different time scales affect drought duration and intensity, as well as the progression of such characteristics in a specific hydrological system. For example, in this study (red) we obtain that, in the Palos River basin, the soil column buffers the intensity of meteorological droughts, which transitions toward a shorter and more intense hydrological drought during the 1998/99 event. Using 1-month (Wan et al., 2018) and 3-month (green; Gautam et al., 2024) scales for SPI, SSMI and SSI yields a transition from a very intense and short meteorological drought towards a longer and smoother hydrological drought; nevertheless, the time scales recommended by Baez-Villanueva et al. (2024, blue) yield a decline in intensity and a slightly shorter duration from meteorological to hydrological drought. Other discrepancies in drought trajectories are obtained in all combinations of basin/event.

Note that the relative location of soil moisture drought within the trajectories can be very different depending on the time scale selected. An interesting example is the 2012-2016 event at the Choapa River basin, for which the four trajectories differ considerably; in particular, the time scales found here yield very similar durations for meteorological and hydrological droughts, and a more intense and prolonged soil moisture drought. Notably, Figure 8 also shows that our trajectories (red symbols and arrows) for the two events analyzed are similar at the Palos and Cautín River basins, suggesting a similar propagation pattern between mixed and rainfall-driven regimes. The same pattern is also obtained for the Ñuble River basin (Figure S3 in the Supplement).

It should be noted that the time scales selected based on maximum correlation with the SSI-6 (or any other aggregation) do not necessarily yield similarities between the onset, duration and end obtained with different indices for an individual event. Figure 9 illustrates this point for two events at the Cautin River basin. The results show that SSMI-12 and SWSI-12 correlate well with the temporal evolution of the SSI-6 for the 1998/99 drought; nevertheless, the temporal variability of SSMI-12 during the 2016/17 event shows a closer agreement with the SSI-1 compared to SSI-6 which, in turn, does not match with the SSMI and SWSI at any aggregation, but yields a similar onset, duration and end detected with the SPI-12 and SPEI-12. Even more, the SSMI and SWSI reflect soil moisture and total storage deficits, respectively, before the precipitation deficits detected with the SPI-12 using a -1 threshold.



## 5 Discussion

### 5.1 Drought detection and characteristics


This study reveals additional insights for hydrological drought analysis based on SSI estimates. Despite the results confirm well-known effects of the temporal scale selected for streamflow aggregation on the frequency and duration of hydrological droughts detected with the SSI (e.g., Barker et al., 2016; Teutschbein et al., 2022), such impacts are minor in slow-reacting catchments (e.g., Cochiguaz River basin, with average drought durations ranging 12.3-12.9 months), which can be explained

by the buffering effect of snowpack, as well as soil moisture and aquifer storage. Conversely, the impacts of the temporal scale and duration constraints are more noticeable in rainfall-driven basins, where considerable rainfall contributions to runoff occur during winter. Note that the relatively longer average drought durations found in semi-arid, snowmelt-driven catchments (which also hold the largest baseflow contributions) align well with previous studies linking drought duration with catchment storage properties (Van Loon and Laaha, 2015; Barker et al., 2016).

### 5.2 Interpretability of the SSI across hydrological regimes


In rainfall-driven basins, we found a strong connection between SSI-6 and precipitation deficits (i.e., a strong link between hydrological and meteorological drought), whereas soil and aquifer storages become more important in basins with increased aridity, in agreement with previous studies (e.g., Haslinger et al., 2014; Barker et al., 2016). Specifically, in semi-arid basins the SSI reflects the variability of >12-month aggregated precipitation and SWE, and fluctuations in aquifer storage at <9-

month timescales. Our results also show that dependencies between correlations and hydroclimatic regimes change with the analysis period (Figure 7), highlighting the uniqueness of each drought event.

We show that aggregating streamflow into seasonal periods (i.e., 3 and 6 months) for SSI calculations does not necessarily attenuate potential relationships with other water cycle variables (e.g., see results for the Cochiguaz River basin, Figure 4). Even more, shifting from SSI-1 to SSI-3 and SSI-6 yields a higher influence of soil moisture and aquifer storage for nearly all

temporal scales in mixed and rainfall-driven regime basins. These results suggest that the time scale used for the SSI should be selected based on the specific purposes and the hydroclimatic regime if the aim is to enhance the interpretability of physical mechanisms.

Although previous studies have shown that meteorological droughts may propagate differently depending on hydroclimatic characteristics and system properties (e.g., Van Loon et al., 2014; Van Loon & Laaha, 2015; Barker et al., 2016; Apurv et al.,

2017), we show that such portrayal may be very sensitive – for a given combination of event and catchment – to the subjective choice of the time scale used to compute standardized indices (Figure 9). Moreover, given a well-defined criterion to compute standardized indices (in this study, SPI-12, SSMI-12, and SSI-6), the trajectories of the same drought event may differ considerably among catchments. Likewise, propagation trajectories can differ substantially among drought events within a particular catchment (Figure S4 of the Supplement). Overall, this work suggests that any results derived from standardized





indices should be interpreted cautiously, checking carefully the reasoning behind the selection of the selected drought indices and their temporal scales.

### 5.3 Implications for operational practice

In Chile, the current legislation states that hydrological droughts between the Atacama and Araucanía Regions – a large area that encloses the six basins examined here – are officially declared based on the SSI-6 (DGA, 2022), ignoring the spatial

diversity in hydroclimatic and physiographic catchment characteristics within that domain. Paradoxically, DGA (2022) considered spatial differences in that area regarding the SPI, recommended a 12-month time scale between the Atacama and Maule Regions (which encloses our snowmelt-driven and mixed regime basins), and a 6-month aggregation between the Ñuble and Araucania Regions (which encloses the two rainfall-driven basins analyzed here).

The results presented here suggest that the choice of time scales for the SSI should be made depending on the hydroclimatic

features of the basin of interest and the target application(s). In this regard, obtained that the temporal scale selected for the SSI is not as relevant in snowmelt-driven basins, compared to mixed regime and rainfall-dominated catchments. For real-time hydrological drought monitoring or to characterize short and intense events, 1 to 6-month time scales may be convenient, whereas ≥12 months would be more suitable for multi-year drought detection, since long time scales help to smooth the original temporal variability and capture the long-term effects of precipitation deficits (Figure 4). If surface water is used for irrigation,

the choice of time scale should also consider the specific crop characteristics and, in particular, the capability (i.e., period length) to survive under water scarcity conditions.

### 5.4 Limitations and future work

In this study, we did not consider in-situ or remotely-sensed observations of SWE, soil moisture, and aquifer storage in the calibration process, relying on the capability of the SUMMA model to replicate streamflow signatures. Moreover, we did not

test different modeling alternatives (e.g., baseflow parameterizations) or spatial configurations (e.g., spatially varying soil layer depths) across basins.

Future work could expand the analyses presented in this study by exploring tradeoffs between the time scales used to compute the SSI and the choice of statistical distributions (e.g., Svensson et al., 2017; Teutschbein et al., 2022), the parameter estimation method (e.g., Tijdeman et al., 2020) the choice of reference period (set here as 1983/84-2012/13), or the threshold selection

criteria (e.g., Wanders et al., 2015; Odongo et al., 2023). Finally, using a large sample of basins (e.g., Vásquez et al., 2021; Muñoz-Castro et al., 2023) would enable designing regional analysis frameworks based on similarities in geology, land cover and hydrology (e.g., Anderson and Schilling, 2024), and draw more robust conclusions for mountainous domains.





## 6 Conclusions

The standardized streamflow index (SSI) has been widely used for hydrological drought monitoring, forecasting, and
propagation analyses. Nevertheless, there is limited understanding of how the subjective choice of time scales affect the characterization of these events and, more importantly, which hydrological variables are related to SSI fluctuations. In this study, we intend to fill these gaps by applying the SUMMA hydrological model coupled with the mizuRoute routing model, in six hydroclimatically different basins located on the western slopes of the extratropical Andes. We also illustrate how sensitive the portrayal of drought propagation is to the time scales used to compute popular standardized indices such as the
SPI and the SSMI. Our main findings are as follows:

1. The time scale used to compute the SSI and the minimum duration to define hydrological drought occurrence can largely affect the estimated duration and frequency of these events, especially in rainfall-driven catchments.

2. The strength of the relationship between the SSI and hydrological storages/fluxes is less affected by the choice of time scale of the SSI in snow-driven regimes compared to mixed and rainfall-dominated basins, where the dispersion
of correlations progressively increases when evaluating the indices on scales larger than nine months.

3. Higher correlations are achieved when SSI-6 is contrasted against hydrological variables aggregated at 9 and 12 months, except for aquifer storage at the Cochiguaz basin (snowmelt-driven), and lower correlations are obtained for time scales longer than 12 months. When the SSI-1 and SSI-3 are used, the correlations are maximized at shorter temporal scales (compared to the SSI-6) for some combinations of hydrological variables and basins (e.g., aquifer
storage at Palos and Ñuble).

4. When analyzing the entire period (April/1983 - March/2020), higher correlations between the SSI-6 and hydrological variables are achieved for snowmelt-driven basins, and these progressively decrease towards rainfall-driven regimes. This pattern becomes stronger when the total water storage within a basin is considered. Nevertheless, such a pattern becomes less clear and dependent on the temporal aggregation of explanatory variables during drought periods.

5. The portrayal of drought propagation may change drastically depending on the choice of time scales used to compute standardized indices. In this regard, different criteria may reveal opposite trajectories of drought propagation for the same event in a basin.

## 7 Data availability

The CR2METv2.0 dataset is available at https://www.cr2.cl/datos-productos-grillados (Boisier et al., 2018). Daily streamflow
records from Chilean Water Directorate (DGA) is available at https://www.cr2.cl/datos-de-caudales/. The ERA5-Land data can be downloaded from https://doi.org/10.24381/cds.e2161bac (Muñoz-Sabater et al., 2021). Land cover data from MODIS MCD12C1 product can be found at https://modis.gsfc.nasa.gov/data/dataprod/mod12.php



## 8 Author contributions

FL, PM and NV conceptualized the study, designed the overall approach and wrote the manuscript. FL conducted all the model
simulations, analyzed the results and created most of the figures. NV provided support to set up the scripts used in this study.
All the authors contributed to refine the methodology and analysis framework, interpretation of results, reviewing and editing
the manuscript.

## 9 Competing interests

The authors declare that they have no conflict of interest.

## 10 Financial support

Fabián Lema, Pablo A. Mendoza, and Nicolás Vásquez were supported by Fondecyt Project 11200142. Pablo A. Mendoza
also received financial support from ANID-PIA Project AFB230001 (AMTC). Nicolás Vásquez received additional support
from the Emerging Leaders in the Americas Program (ELAP) scholarship (Canada) and the ANID Doctorado Nacional
scholarship No. 21230289 (Chile). Mauricio Zambrano-Bigiarini acknowledges support from the FONDAP Research Center
(CR)2 (15110009) and from ANID Fondecyt 1212071.

Powered@NLHPC: This research was partially supported by the supercomputing infrastructure of the NLHPC (ECM-02).

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





**Table 1. Physiographic and climatic attributes of the six basins considered in this study. All data came from the CAMELS-CL database, except for the baseflow index, which was estimated from hydrological simulations in the SUMMA model. The aridity index was calculated as PET/P.**

| Catchment | ID | Lat. (°) | Long. (°) | Elevation range (m) | Area (km²) | Mean annual P (mm/yr) | Mean anual Q (mm/yr) | Runoff ratio (-) | Aridity index (-) | Baseflow index (-) |
|---|---|---|---|---|---|---|---|---|---|---|
| Cochiguaz | 4313001 | -30.30 | -70.28 | 1341-5275 | 675 | 259 | 114 | 0.44 | 3.8 | 0.99 |
| Choapa | 4703002 | -32.10 | -70.45 | 1153-5038 | 1132 | 392 | 231 | 0.59 | 2.3 | 0.98 |
| Claro | 6027001 | -34.85 | -70.73 | 542-3046 | 349 | 1414 | 891 | 0.63 | 0.7 | 0.42 |
| Palos | 7115001 | -35.44 | -70.74 | 590-3282 | 490 | 1960 | 1686 | 0.86 | 0.5 | 0.81 |
| Ñuble | 8105001 | -36.68 | -71.19 | 645-3189 | 1254 | 2108 | 1792 | 0.82 | 0.5 | 0.71 |
| Cautín | 9123001 | -38.47 | -71.75 | 413-3090 | 1306 | 2906 | 2092 | 0.72 | 0.4 | 0.72 |


**Table 2. List of recent drought propagation studies, including time scales used or recommended for computing standardized drought indices.**

| Drought propagation study | Time scales used (months) | | |
|---|---|---|---|
| | SPI | SSMI | SSI |
| Barker et al. (2016) | 1, 6 and 18 | - | 1, 6 and 18 |
| Huang et al. (2017) | 1 and 6 | - | 1 |
| Wu et al. (2017) | 3 | - | 3 |
| Wan et al. (2018) | 1 | 1 | 1 |
| Peña-Gallardo et al. (2019) | 1-48 (SPI/SPEI) | - | 1 |
| Bhardwaj et al. (2020) | 1 (3-month smoothed) | 1 (3-month smoothed) | 1 (3-month smoothed) |
| Fuentes et al. (2022) | 3 (SPI/SPEI) | 3 (SVI*) | 3 (SRI*) |
| Odongo et al. (2023) | 1-9 | 1 | 1 |
| Adeyeri et al. (2023) | 3 | - | 3 (SRI*) |
| Gautam et al. (2024) | 3 | 3 | 3 |
| Baez-Villanueva et al. (2024) | 12-24 (nival) 3-12 (nivo-pluvial) 3-6 (pluvial) | 6-12 (nival) 1-3 (nivo-pluvial) 1-3 (pluvial) | 1 |
| This study | 12 | 12 | 6 |

Note: SVI refers to Standardized Vegetation Index, calculated based on the MODIS NDVI index and described in Peters et al. (2002). SRI refers to the Standardized Runoff Index (Shukla and Wood, 2008).






**Table 3. Mean duration (in months) of drought events for each case and temporal scale of SSI.**

| Basin | Case 1: free | | | Case 2: constrained | | |
|---|---|---|---|---|---|---|
| | SSI-1 | SSI-3 | SSI-6 | SSI-1 | SSI-3 | SSI-6 |
| Cochiguaz | 12.25 | 12.50 | 12.88 | 12.25 | 12.50 | 12.88 |
| Choapa | 4.58 | 6.60 | 8.67 | 8.27 | 7.39 | 9.36 |
| Claro | 2.58 | 4.50 | 6.20 | 5.80 | 6.15 | 7.39 |
| Palos | 3.50 | 6.46 | 7.77 | 6.33 | 8.00 | 9.00 |
| Ñuble | 3.04 | 4.50 | 5.28 | 5.90 | 7.27 | 7.91 |
| Cautín | 2.64 | 3.81 | 6.31 | 6.13 | 6.50 | 8.67 |







Figure 1. Location and seasonality of precipitation (P), runoff (Q) and temperature for the six case study basins: (a) Cochiguaz River at El Peñón, (b) Choapa River at Cuncumén, (c) Claro River at El Valle, (d) Palos River at Colorado, (e) Ñuble River at La Punilla, and (f) Cautín River at Rari-Ruca. Overlines represent annual averages for the period April/1985-March/2015.





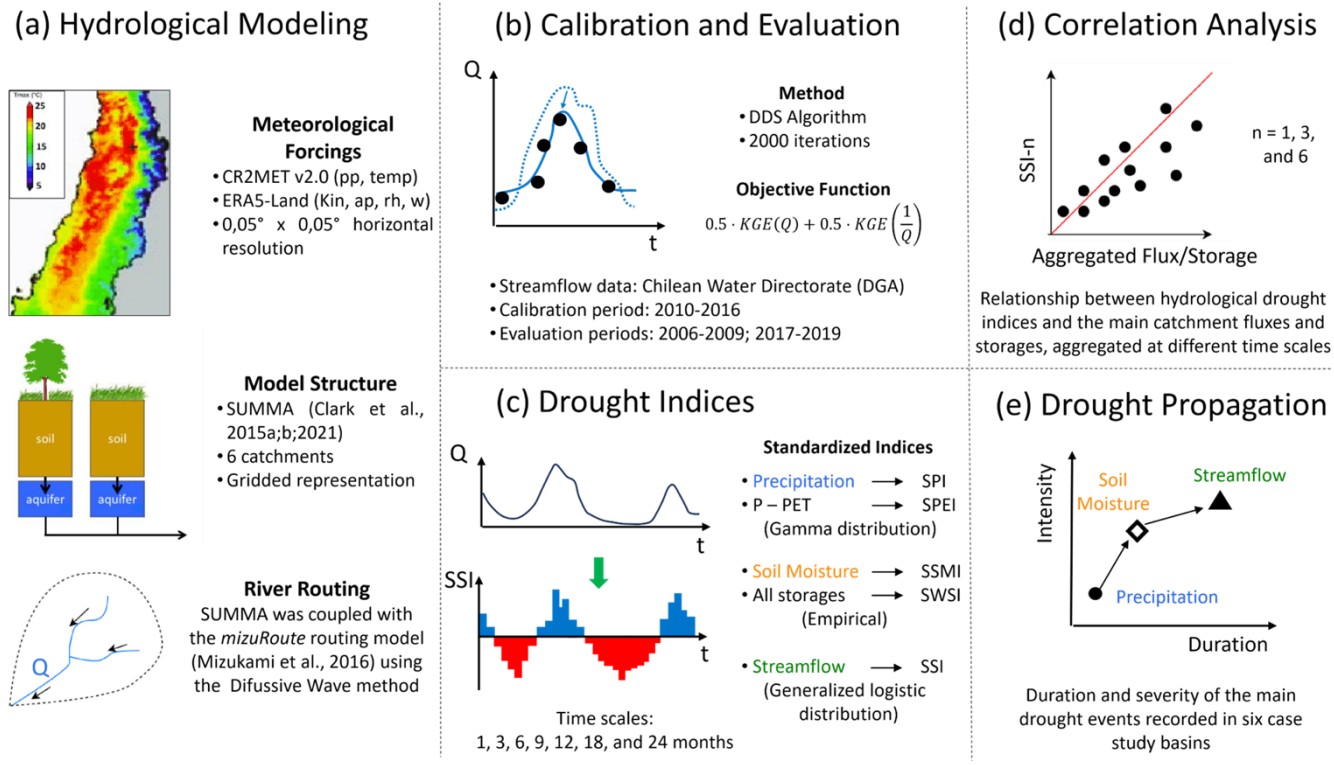

**Figure 2.** Flowchart describing the approach used in this study. See text for details.






**Figure 3. Comparison between simulated and observed streamflow for all basins in terms of (a) daily time series (April/2014 to March/2020), (b) daily flow duration curves (vertical logarithmic scale), and (c) mean monthly runoff. In (a) the shaded area represents part of the calibration (yellow) and evaluation (white) periods, and OF indicates the value of the objective function (Eq 3.1) over the evaluation period. The results in (b) and (c) correspond to the evaluation periods (April/2006 – March/2010 and April/2017 – March/2020) combined.**







**Figure 4. Monthly time series of (a) precipitation, (b) SPI, (c) SWE, (d) total soil moisture, (e) aquifer storage, (f) streamflow, and (g) SSI for the Choapa (snowmelt-driven, left) and Cautín (rainfall-driven, right) River basins. Monthly precipitation and SPI-n are obtained from the CR2MET meteorological product, whereas the remaining variables are obtained from SUMMA model simulations during January/1998-December/2000. Time scales of 1, 3 and 6 months for the SSI are highlighted due to their widespread use (see text for details).**





**Figure 5. Effects of the choice of temporal scale (1, 3, and 6 months) in SSI calculations and duration restrictions on (a) the frequency and (b,c) duration of hydrological droughts detected between the water years 1983/84-2019/20. Probability distributions of drought durations are displayed for the cases (b) no restrictions (i.e., "free"), and (c) minimum drought duration of three months (i.e., "constrained") for event detection (see text for details).**





**Figure 6. Spearman rank correlation coefficients between the SSI computed at different time scales (1, 3, and 6 months), and temporally aggregated/averaged hydrological variables (columns) over the period Jan/1998-Dec/2000. The results for each case study basin are displayed in different rows.**







**Figure 7. Spearman rank correlation coefficients between SSI-6 and temporally aggregated/averaged catchment-scale hydrological variables (rows) for three different periods: the October/1998-September/1999 drought event, (b) the central Chile megadrought (April/2010-March/2019), and (c) the entire analysis period (April/1983 - March/2020).**



**Figure 8. Propagation from meteorological (circles) to soil moisture (diamonds) and hydrological (triangles) droughts
for two selected drought events (1998/99 and 2012/-2016, displayed in different rows) and three basins with different
hydrological regimes: (a) Choapa (snowmelt-driven, left), (b) Palos (mixed regime, center) and Cautín (rainfall-driven,
right). The x-axis shows the duration in months, and the y-axis displays the intensity. The colors indicate trajectories
obtained with the temporal scales recommended by different studies (see text for details).**

815

820

825



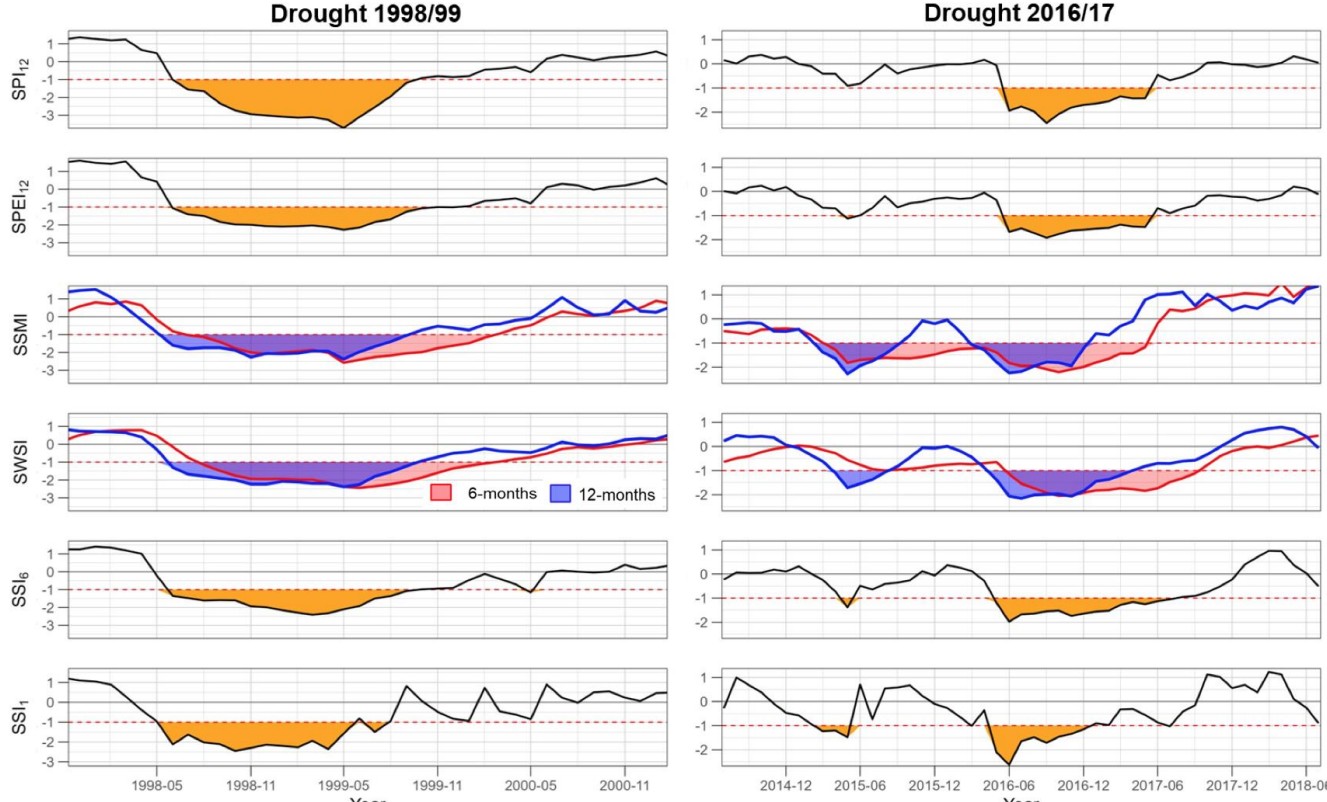

**Figure 9. Monthly time series of standardized indices at the Cautín River basin, computed with the time scales selected from the correlation analyses. The orange areas illustrate the onset, end, and duration of the 1998/99 drought (left), and the 2016/17 drought (right), according to the different indices. For the SSMI and the SWSI, two time scales (6-month and 12-month values) are displayed for comparison (see text for details).**