# Peer review of "Technical note: What does the Standardized Streamflow Index actually reflect? Insights and implications for hydrological drought analysis"

_Hydrology and Earth System Sciences, 2024_

## Author Comment (AC1)

Replies to reviews

**"Technical note: What does the Standardized Streamflow Index actually reflect? Insights and implications for hydrological drought analysis**

Fabián Lema, Pablo A. Mendoza, Nicolás Vásquez, Naoki Mizukami, Mauricio Zambrano-Bigiarini, and Ximena Vargas

*We provide responses to each individual point below. For clarity, comments are given in italics, and our responses are given in plain blue text.*

**Anonymous Referee #1**

*The paper demonstrates strong scientific significance, high quality, and effective presentation. The authors investigate hydrological droughts, focusing on Standardized Drought Indices (SDI), notably the Standardized Streamflow Index (SSI), as tools for understanding drought dynamics, including frequency, intensity, duration, and propagation. The SUMMA hydrological model and the mizuRoute routing model were calibrated and used to analyze six case study basins in the western extratropical Andes. This analysis explores the relationship between SSI and various explanatory basin-aggregated variables, such as precipitation and catchment storage, across different time scales.*

*The authors address the common use of SSI to quantify hydrological drought without clearly understanding its effectiveness in capturing the dynamics of drought propagation in basins with diverse hydrological regimes. This key issue is woven throughout the paper, making it enjoyable to read.*

*Although the authors work with a limited number of basins, their analysis has implications beyond the extratropical Andean region, particularly in operational contexts where regulators and decision-makers rely on simplified and easily accessible drought indices without further analysis or differentiation between hydrological regimes. The discussion section provides a solid overview of the approach's limitations and potential avenues for future research. The figures convey important information, making it easy to grasp the main findings.*

We thank the referee for meticulously reviewing our manuscript and providing several constructive suggestions. We are especially grateful for the referee's positive feedback.

*Minor comments:*

1. *While the justification for using a hydrological model as a benchmark for evaluating SDIs over observational data is clear, I believe the paper would benefit from a more detailed discussion of the inherent limitations of using a model to represent the long-term behavior of drought in the chosen basins. Analyzing how the model's process representation might lead to inaccuracies in streamflow reproduction—especially in basins where the modeled minimum streamflow values exceed the observed values— could provide additional insight into the analysis.*

We have added the following text in section 4.1 regarding inaccuracies in low flow simulations (L257-L261):

"… there is an overestimation of low flow volumes with exceedance probabilities larger than 90% in the Choapa and Claro catchments (< 2 $m^3$/s), which could be explained by the inadequate model physics representation including, but not limited to the lack of a common aquifer enabling water exchange among grid cells in our SUMMA configuration, and/or biases in the forcing dataset that impact the accumulation and melting of snow."

In response to this comment and another reviewer's observation, we have added the following text in section 5.1 (L377-L382):

"Although the model's overestimation of low flow volumes in Choapa and Claro (Figure 3) affects the accuracy (i.e., closeness to reality) of the number and duration of detected events (Figure 5), this artifact does not alter our conclusions, as all analyses focus on the impact of methodological choices related to index calculations using simulated variables, regardless of the fidelity of model representations. Even more, all the correlation and drought propagation analyses were performed in the model's world and, therefore, streamflow biases should not impact the extent to which variables or drought indices computed with different time scales relate to each other".

2. *I suggest the authors include more information regarding the model's warm-up period or restrict the time results of their full simulation to account for this warm-up period.*

We did consider a spin-up period before computing all the performance metrics for hydrological model calibration and evaluation. We have clarified this point in L191-L193:

"The observed daily streamflow data is split into a warm-up period (April/2004 – March/2006), a calibration period (April/2010 – March/2017), and two non-consecutive evaluation periods (April/2006 – March/2010 and April/2017 – March/2020)."

We also consider a two-year warm up period before computing the standardized drought indices. We clarify this point in L217-L219:

"We use the calibrated parameters (see section 3.2) to perform hydrologic simulations for the historical period April/1981 – March/2020. All SDI computations consider a spin-up period of two years (April/1981 – March/1983) and the same reference period of 30 years (April/1983 – March/2013)".

3. *Lastly, I would appreciate it if the authors could elaborate on how they envision the design of regional analysis frameworks that consider more than just hydrological regimes, including similarities in physical features such as slope, elevation, soil properties, and land cover, among others.*

This is an interesting point. The analyses presented here could be expanded to a larger number of basins that consider a greater diversity of features (e.g., Vásquez *et al.*, 2021,

Muñoz-Castro *et al.*, 2023), in order to examine whether the time scales of hydrological variables (e.g., precipitacion, soil moisture, SWE) that maximize the correlation (or 'optimal' time scales) with the SSI are related to physiographic attributes such as contributing area, slope, elevation, geology, land cover and soil type, among others. A simple stratification of attribute values by optimal time scale, or any other hydrological descriptor of interest (e.g., Sawicz *et al.*, 2011, Almagro *et al.*, 2024) could provide valuable insights, complementing previous drought investigations using large samples of catchments. For example, Van Loon and Laaha (2015) found that geology and land use were relevant controls for hydrological drought duration. Peña-Gallardo et al. (2019) concluded that elevation and vegetation coverage are the main factors controlling the diverse response of SSI to SPEI time scales. More recently, Brunner and Stahl (2023) confirmed that land surface processes are required to explain the temporal clustering of hydrological droughts. More generally, additional large-sample hydrology analyses could help to improve our understanding of the main drivers affecting drought occurrence and propagation across different hydroclimates. We will incorporate these ideas in the Discussion section (L447-L457).

*L235-L245: Can the authors discuss the implications of using a model that overestimates the low flow volumes (exceedance probabilities over 90%) for analyzing hydrological droughts for the Choapa and Claro cases?*

We thank the reviewer for making this point. Although the model' s overestimation of low flow volumes in Choapa and Claro (Figure 3) affects the accuracy (i.e., closeness to reality) of the number and duration of detected events (Figure 5), this artifact does not alter our conclusions, as all analyses focus on the impact of methodological choices related to index calculations using simulated variables, regardless of the fidelity of model representations. Even more, all the correlation and drought propagation analyses were performed in the model's world and, therefore, streamflow biases should not impact the extent to which variables or drought indices computed with different time scales relate to each other. We will clarify these points in section 5.1 ("Drought detection and characteristics") of the revised manuscript (L377-L382).

*Additionally, what are the most likely causes of the model's misrepresentation of low flow volumes in those basins? Can some of the selected process parameterizations (i.e., snowmelt) negatively affect the obtained results in this respect?*

We speculate that both model structural deficiencies and forcing errors may be the main causes of the misrepresentation of low flow volumes. The SUMMA configuration used in this study considers that precipitation is the only water input for each grid cell, and lateral water exchanges among modeling units – in particular, groundwater fluxes – are not allowed, which could explain the relatively lower performance in the Cochiguaz and Claro River basins. On the other hand, CR2MET v2.0 – which is the baseline dataset for model simulations – can be considered a bias correction of daily ERA5 precipitation and extreme temperature outputs. Because such correction relies on meteorological gauges, which are very sparse in the Andes Cordillera, the resulting forcing dataset may contain biases that affect the accumulation and melting of snow, both extremely relevant in the Cochiguaz and Claro River basins since they are snowmelt-driven catchments. To reflect these points, we have modified the text as follows (L257-L261):

"… there is an overestimation of low flow volumes with exceedance probabilities larger than 90% in the Choapa and Claro catchments (< 2 m³/s), which could be explained by the lack of a common aquifer enabling water exchange among grid cells in our SUMMA configuration, and/or biases in the forcing dataset that impact the accumulation and melting of snow."

*Also, did the authors evaluate the influence of multiple model parameterizations on the obtained results?*

We did not explore the effects of using multiple parameterizations on the results and conclusions and, therefore, we will add the following text to section 5.4 (L439-L441):

"We did not explore the effects of using alternative model parameterizations (e.g., stomatal resistance, lateral fluxes) or spatial configurations (e.g., spatially varying soil layer depths) on the results and conclusions obtained."

*L260-L276: Did the authors consider a spin-up period for the model before starting the analysis in 04/1983? If so, please include this information. If not, I'd recommend neglecting the first two simulation years (1983-1984) in the subsequent analysis to minimize the influence of initial conditions in the obtained results.*

We did consider a two-year warm up period before computing the standardized drought indices. We clarify this point in L217-L219:

"We use the calibrated parameters (see section 3.2) to perform hydrologic simulations for the historical period April/1981 – March/2020. All SDI computations consider a spin-up period of two years (April/1981 – March/1983) and the same reference period of 30 years (April/1983 – March/2013)".

*L36: "associated to" replace with "associated with."*

We will modify the text following the reviewer's recommendation.

*L42: "Despite the drought concept refers" replace with "Despite the drought concept referring."*

We will modify the text following the reviewer's recommendation.

*L80: "percentile-based thresholds that are commonly" replace with "percentile-based thresholds commonly."*

We will modify the text following the reviewer's recommendation.

*L92: "What are the effects of different time scales on" replace with "How do different time scales affect"*

We will modify the text following the reviewer's recommendation.

*L94: "towards" replace with "toward".*

We will modify the text following the reviewer's recommendation.

*L96: "To seek for answers," replace with "To seek answers,"*

We will modify the text following the reviewer's recommendation.

*L113: "Hereafter, to" replace with "Hereafter,"*

We will modify the text following the reviewer's recommendation.

*L117 & L118: "mean annual temperatures between 9 to 16 °C" and "aridity indices between 0.4 to 3" replace with "mean annual temperatures between 9 and 16 °C" & "aridity indices between 0.4 and 3".*

We will modify the text following the reviewer's recommendation.

*L349: "drought durations ranging 12.3-12.9 months" replace with "drought durations ranging from 12.3-12.9 months"*

We will modify the text following the reviewer's recommendation.

*L386: "SSI is not as relevant in snowmelt-driven basins, compared to mixed regime and rainfall-dominated catchments." This statement appears weak. I recommend replacing it with: "SSI is less relevant in snowmelt-driven basins than in mixed regimes and rainfall-dominated catchments."*

We will modify the text following the reviewer's recommendation.

**References**

Almagro, A., Meira Neto, A. A., Vergopolan, N., Roy, T., Troch, P. A. & Oliveira, P. T. S. (2024) The Drivers of Hydrologic Behavior in Brazil: Insights From a Catchment Classification. *Water Resour. Res.* **60**(8). doi:10.1029/2024WR037212

Brunner, M. I. & Stahl, K. (2023) Temporal hydrological drought clustering varies with climate and land-surface processes. *Environ. Res. Lett.* **18**(3). doi:10.1088/1748-9326/acb8ca

Loon, A. Van & Laaha, G. (2015) Hydrological drought severity explained by climate and catchment characteristics. *J. Hydrol.* **526**, 3–14. Elsevier B.V. doi:10.1016/j.jhydrol.2014.10.059

Muñoz-Castro, E., Mendoza, P. A., Vásquez, N. & Vargas, X. (2023) Exploring parameter (dis)agreement due to calibration metric selection in conceptual rainfall-runoff models. *Hydrol. Sci. J.* Taylor & Francis. doi:10.1080/02626667.2023.2231434

Peña-Gallardo, M., Vicente-Serrano, S. M., Hannaford, J., Lorenzo-Lacruz, J., Svoboda,

M., Domínguez-Castro, F., Maneta, M., et al. (2019) Complex influences of meteorological drought time-scales on hydrological droughts in natural basins of the contiguous Unites States. *J. Hydrol.* **568**(November 2018), 611–625. Elsevier. doi:10.1016/j.jhydrol.2018.11.026

Sawicz, K., Wagener, T., Sivapalan, M., Troch, P. A. & Carrillo, G. (2011) Catchment classification: empirical analysis of hydrologic similarity based on catchment function in the eastern USA. *Hydrol. Earth Syst. Sci.* **15**(9), 2895–2911. doi:10.5194/hess-15-2895-2011

Vásquez, N., Cepeda, J., Gómez, T., Mendoza, P. A., Lagos, M., Boisier, J. P., Álvarez-Garretón, C., et al. (2021) Catchment-Scale Natural Water Balance in Chile. In: *Water Resources of Chile*, 189–208. doi:10.1007/978-3-030-56901-3_9

---

## Author Comment (AC2)

Replies to reviews

**"Technical note: What does the Standardized Streamflow Index actually reflect? Insights and implications for hydrological drought analysis**

Fabián Lema, Pablo A. Mendoza, Nicolás Vásquez, Naoki Mizukami, Mauricio Zambrano-Bigiarini, and Ximena Vargas

We provide responses to each individual point below. For clarity, comments are given in italics, and our responses are given in plain blue text.

**Anonymous Referee #2**

*The manuscript provides a detailed investigation into the Standardized Streamflow Index (SSI), a widely used metric for characterizing hydrological drought. The authors employ the SUMMA hydrological model coupled with the mizuRoute routing model across six diverse basins in the Andes to explore the relationships between SSI and potential explanatory variables. Their analysis extends to the impacts of time-scale selection on drought propagation, offering insights into the complex dynamics of drought characterization. The study emphasizes the importance of cautious selection of drought indices and time scales, highlighting their influence on event characterization, monitoring, and propagation analysis.*

*While the study is timely and relevant, there are areas where the manuscript could be improved to enhance its clarity, rigor, and accessibility. A thorough revision will make the manuscript more accessible to a broader audience and enhance its scientific impact.*

We thank this reviewer for his/her time in commenting on our paper. We have addressed all the comments provided by this reviewer. Please see our individual responses below.

1. *Clarity and Consistency*
*The manuscript often lacks consistency in the use of acronyms and terminology. For instance, precipitation is referred to as "pp" and "P" (e.g., Figure 2). This inconsistency can be confusing to readers. Ensure all acronyms are concise, clearly defined, and consistently used throughout the text and figures. Additionally, figures should "stand-alone" with comprehensive captions that explain all acronyms and variables.*

We thank this reviewer for thoroughly revising our manuscript. In response to this comment, we have revised the use of all acronyms throughout the paper, and have defined them accordingly to avoid confusion among readers. We have also rewritten the caption of Figure 2, in order to define to include the definition of all acronyms and variables referred in this figure.

2. *Methodology Organization*

*The methods section is repetitive and lacks a clear structure. For instance, the "Approach" subsection is confusing and does not align with Figure 2 or the steps described later. Additionally, some methodological details are scattered throughout the manuscript or*

*mixed with results (e.g., lines 263–265). Reorganizing this section for clarity and separating methods from results would greatly improve readability.*

In response to this comment, we have moved methodological descriptions from section 2 to section 3.1. We have also moved the following description, originally located in subsection 4.2 (Results), to subsection 3.3 (Approach), in order to keep focus and clarity (L221-L224):

"To this end, we apply a fixed threshold criterion (Van Loon, 2015) – set here as -1 – in two different ways: (i) a drought event starts when SDI-n drops below -1 and ends when it reaches or exceeds -1 – i.e., it is possible to detect one-month events ("free" criteria) –; and (ii) a drought event begins when SDI-n remains below -1 for at least three consecutive months and concludes when it reaches or exceeds -1 ("constrained" criteria)."

We have rewritten the caption of Figure 2 and re-organized the subsections contained in section 3 (Approach), re-numbering from 3.1 to 3.5 –, to connect the methodological steps illustrated in Figure 2 with their respective subsection. Lastly, we have revised the first paragraph of Section 3 to improve the connection between subsequent methodological descriptions and Figure 2.

3. *Time-Scale Terminology*

*The manuscript uses "time scales" to refer to different concepts—indices aggregation periods and aggregated flux/storage—without clear differentiation. This ambiguity makes the text hard to follow. Clearly define these terms early in the methods section and ensure consistent usage throughout.*

We thank the reviewer for this comment. In this paper, we use the terms "time scale" or "temporal scale" when referring to the temporal window used to aggregate (or average) monthly values. For example, the 3-month time scale for September 2015 precipitation is the aggregation of monthly amounts (in mm/month) for July to September 2015. For the case of state variables (e.g., SWE, soil moisture) or fluxes (e.g., streamflow) the 3-month time scale is obtained by averaging monthly means. We clarify this point in section 3 of the revised text of the manuscript (L140-L144).

Additionally, we have removed any reference to "aggregation", "aggregation period" or "temporal aggregation" from the revised manuscript to avoid confusion among readers.

4. *Depth of Analysis*
*While the study highlights interesting patterns, some key analyses, such as drought propagation (lines 82–84), are underexplored in the results and discussion. Either expand on this analysis or remove it to maintain focus and coherence.*

In response to this comment, we have expanded section 4.4 to provide more insights from the results presented in Figure 8 (L336-L436):

"For example, the results for the 1998/99 event in the Choapa River basin show that using 1-month (purple; Wan et al., 2018), 3-month (green; Gautam et al., 2024) and the time scales derived here yield a transition toward a relatively longer and more intense hydrological drought, compared to the meteorological drought, whereas the time scales recommended by Baez-Villanueva et al. (2024, blue) provide a progression toward a more intense and slightly shorter hydrological drought. In the Palos River basin we obtain that, for the same event and the time scales derived from this study (red), the soil column buffers the intensity of the meteorological drought, which transitions toward a shorter and more intense hydrological drought during the 1998/99 event. Using 1-month and 3-month time scales for SPI, SSMI and SSI yields a transition from a very intense and short meteorological drought towards a longer and smoother hydrological drought; nevertheless, the time scales recommended by Baez-Villanueva et al. (2024, blue) yield a decline in intensity and a slightly shorter duration from meteorological to hydrological drought. In the Cautín River basin, all propagation trajectories obtained for same event are very different."

To complement the analyses presented for Figure 8, we have added the following text in section 4.4 (L351-L354):

"For the same event, 1-month (purple), 3-month (green) and the temporal scales from Baez-Villanueva et al. (2024) yield trajectories with decreasing intensity and longer durations as moving from meteorological to soil moisture and hydrological droughts in the Cautín River basin; however, the time scales derived from our analyses (red) indicate a longer soil moisture drought in comparison with the resulting hydrological drought."

Since our drought propagation results reveal potential pitfalls in use of standardized indices, we have expanded the discussion section as follows (L400-L405):

"Further, the results presented here reveal pitfalls in drought propagation analyses when selecting time scales for standardized indices based on correlation analyses and fixed thresholds. Specifically, the results in Figure 9 for the 2016/17 event suggest that, given a drought event affecting a unique hydrological system, the thresholds for standardized meteorological and soil moisture indices that enable interpreting causality in time (including onset, duration and end) may differ, and variable threshold approaches (e.g., Van Loon & Laaha, 2015, Odongo et al., 2023) may be more appropriate to this end."

Regarding the connections between the correlations and the hydroclimatic regimes of the basins, we have added the following text to section 4.3 (L320-L321):

"Such relationship between the strength of the correlations and the hydroclimatic regime are also obtained for the SSI-3 (Figure S2) and, to a greater extent, for the SSI-1 (Figure S1)."

Finally, we have reworded the second paragraph of section 5.2 as follows:

"We show that aggregating streamflow into seasonal periods (i.e., 3 and 6 months) for SSI calculations does not necessarily attenuate potential relationships with other variables of the water cycle (e.g., see results for the Cochiguaz River basin, Figure 4). Even more, shifting from SSI-1 to SSI-3 and SSI-6 yields a stronger influence of soil moisture and aquifer storage

for nearly all temporal scales in mixed and rainfall-driven regime basins. On the other hand, shifting from the SSI-6 to SSI-3 and SSI-1 exacerbates the connections found between the strength of the correlations and the hydroclimatic regime of the basin analyzed. These results suggest that the time scale used for the SSI should be selected based on the specific purposes and the hydroclimatic regime if the aim is to enhance the interpretability of physical mechanisms."

5. *Practical Implications*
*The discussion section, particularly "Implications for operational practices," is engaging and provides valuable insights. Expanding this section with concrete recommendations or case studies would significantly enhance the manuscript's impact.*

We thank the reviewer for this comment and we agree that including recommendations and case studies would increase the manuscript's impact. In view of this, we have decided to add the following text to section 5.3 (L418-L428):

"In other international agencies, it is common practice to use multiple indicators for drought monitoring and early warning systems (Bachmair *et al.*, 2016), rather than relying only on standardized indices such as the SPI and SSI. These indicators often include satellite products and variables simulated by hydrological models, which aligns with the recommendations outlined in the WMO's Handbook of Drought Indicators and Indices (Svoboda & Buchs, 2016). In particular, the European Drought Observatory (EDO) uses the Combined Drought Index (CDI; Sepulcre-Canto *et al.*, 2012), which simultaneously considers three types of indicators: the SPI, the anomalies of simulated soil moisture in the LISFLOOD hydrological model (De Roo *et al.*, 2000), and anomalies of the Fraction of Absorbed Photosynthetically Active Radiation (FAPAR; Gobron *et al.*, 2005). The former is derived from the MOD15A2H satellite product, and is related with vegetation growth and crop productivity. Similarly, the United States Drought Monitor (Svoboda *et al.*, 2002) combines the Palmer Drought Severity Index (PDSI; Palmer, 1965), the SPI, and soil moisture and streamflow percentile-based indicators in their evaluations."

*Specific Comments*

*Line 37: The computation indices are not for the variables "precipitation, simulated soil moisture, and simulated streamflow" but rather for the different drought types that use these variables as inputs. Rephrase for clarity and conciseness.*

In response to this comment, we have modified the text as follows (L42-L44):

"Despite the drought concept referring to the notion of below-average water fluxes and/or storages (Tallaksen & Van Lanen, 2004; Van Loon, 2015), there are several drought definitions and classifications, being meteorological, agricultural (also referred to as soil moisture drought; e.g., Thober et al., 2015; Cook et al., 2018), hydrological (surface and groundwater level deficits), and socioeconomic the most used drought types (Wilhite & Glantz, 1985)."

*Lines 42–45: The phrase "the most commonly used types" is vague and should specify types of what (e.g., drought indices). Reformulate for clarity.*

To clarify the sentence, we have replaced "the most commonly used types" with "the most used drought types" (see previous response).

*Line 55: The statement "the number 163 journal articles" is irrelevant, the number 163 does not have a meaning by itself. Focus instead on the themes or findings from these articles related to SSI and drought.*

In response to this comment, we have removed he aforementioned sentence from the revised manuscript.

*Line 74: Be specific about "even longer" time scales—e.g., 12 or 24 months.*

We refer to 12 and 24 months with "longer time scales". To clarify this point, we have modified the text as follows (L69-L73):

"Because the SSI-1 may be susceptible to short-term fluctuations, other authors have preferred smoothed (e.g., 3-month averages) time series of SSI-1 (e.g., Bhardwaj *et al.*, 2020), 3-month (e.g., Núñez *et al.*, 2014, Wu *et al.*, 2017, Rivera *et al.*, 2021, Adeyeri *et al.*, 2023, Yun *et al.*, 2023), 6-month (e.g., Seibert *et al.*, 2017, Oertel *et al.*, 2020), or even longer (e.g. 12 and 24 months, Teutschbein et al., 2022; Fowé et al., 2023) time scales."

*Line 77: Mention common meteorological drought indices to provide context for readers less familiar with this field.*

In response to this comment, we now provide a couple of examples of popular meteorological drought indices, and hence the new text reads as follows (L63-L67):

"Most drought propagation analyses seek possible relationships between meteorological drought indices such as the Standardized Precipitation Index (SPI; McKee *et al.*, 1993) and the Standardized Precipitation Evapotranspiration Index (SPEI; Vicente-Serrano *et al.*, 2010) – computed for various time scales – and the SSI for some time scale, being one month (SSI-1) the common choice (e.g., Huang *et al.*, 2017, Peña-Gallardo *et al.*, 2019, Stahl *et al.*, 2020, Wang *et al.*, 2020, Wu *et al.*, 2022, Zhang *et al.*, 2022, Odongo *et al.*, 2023, Baez-Villanueva *et al.*, 2024)."

*Line 88: The phrase "we depart from previous hydrological drought" needs clarification. Specify what you mean by "previous" or cite relevant studies*

With "previous hydrological drought assessments" we refer to those studies using a single time scale for the SSI. Therefore, we have modified the text as follows (L85-L86):

"Here, we depart from previous hydrological drought assessments that used a unique time scale for the SSI (e.g., Stahl *et al.*, 2020, Tijdeman *et al.*, 2020, Wu *et al.*, 2022, Baez-Villanueva *et al.*, 2024)…"

*Lines 135: Add the reference to the model used in this study.*

We have added the references of the models used, following the reviewer's recommendation (L133-L134):

"Our approach considers the configuration of the SUMMA hydrological model (Clark et al., 2015a, 2015b) and the mizuRoute routing model (Mizukami et al., 2016, 2021, Figure 2a)".

*Lines 138–139: Explain how you are comparing the different time scales. What criteria or statistical approaches are being applied?*

To clarify how we compare the effects of selected time, we have modified the text as follows (L133-L140):

"Our approach considers the configuration of the SUMMA hydrological model (Clark et al., 2015a, 2015b) and the mizuRoute routing model (Mizukami et al., 2016, 2021, Figure 2a); the calibration and evaluation of the SUMMA model parameters (Figure 2b); the computation of standardized drought indices (SDIs) for precipitation, simulated soil moisture and simulated streamflow, and the examination of time scale effects on hydrological drought frequency and duration (Figure 2c, see details in section 3.3); and correlation analysis between the SSI and other simulated hydrological variables (Figure 2d). Finally, we examine how time scales typically adopted for the calculation of standardized indices affect the portrayal of historically observed drought events (Figure 2e); specifically, we analyze the transitions from meteorological to soil moisture and hydrological droughts in the duration-intensity space (see details in Section 3.5)".

*Lines 143–145: Define acronyms again for clarity and specify what is meant by "other indices" and "state variables."*

We have defined the acronyms, removed "other indices" and provided some examples for state variables. Therefore, the text has been modified as follows (L146-L152):

"This approach departs from previous efforts searching for statistical relationships between the SSI – computed with streamflow observations – and standardized indices such as the Standardized Precipitation Index (SPI; e.g., Barker *et al.*, 2016, Huang *et al.*, 2017, Wu *et al.*, 2022), the Standardized Precipitation and Evapotranspiration Index (SPEI; e.g., Peña-Gallardo *et al.*, 2019, Wang *et al.*, 2020, Bevacqua *et al.*, 2021), the Standardized Soil Moisture Index (SSMI; Carrão et al., 2013), or other indices and state variables (e.g., soil moisture, aquifer storage, SWE, total water storage) derived from reanalysis datasets that do not necessarily correspond to observed streamflow anomalies (e.g., Hoffmann *et al.*, 2020, Baez-Villanueva *et al.*, 2024)."

*Dataset Description: Add a table summarizing the spatial and temporal resolution of each data product, along with sources and citations. Begin with the variables required for the*

*model, then delve into dataset specifics. Clarify whether streamflow and catchment characteristics were retrieved from CAMELs.*

In response to the reviewer's recommendation, we have included an additional table (Table 2), which is displayed below, summarizing the horizontal and temporal resolutions of each dataset used in the study. Additionally, we have incorporated the following text (L130-L131):

"Table 2 provides a summary of the datasets used in this study, including their horizontal and temporal resolutions."

**Table 2. Datasets used in this study.**

| Variable | Dataset | Horizontal resolution | Temporal resolution | Authors |
|---|---|---|---|---|
| Precipitation and extreme daily temperatures | CR2MET v.2.0 | 0.05° x 0.05° | Daily | DGA, 2017; Boisier et al., 2018 |
| Wind speed, incoming shortwave radiation, atmospheric pressure, and relative humidity | ERA 5-Land | 0.1° x 0.1° | 3-hours | Muñoz-Sabater et al., 2021 |
| Land cover | MODIS MCD12C1 | 0.05° x 0.05° | Yearly | National Aeronautics and Space Administration (NASA) |
| Catchment attributes | CAMELS-CL | - | - | Alvarez-Garreton et al., 2018 |
| Streamflow records | Chilean Water Directorate (DGA) records | - | Daily | Chilean Water Directorate (DGA) |

*Model Calibration and Evaluation: Specify the time scale used for model calibration (e.g., daily or monthly). Additionally, explain the number of trials conducted and the rationale for using the objective function proposed by Garcia (line 175).*

The calibration objective function was selected because it provides a good compromise to achieve good high flow and low flow simulations (Garcia *et al.*, 2017). The metric is computed using daily time series of observed and simulated Q and 1/Q, where Q is streamflow. Finally, we set a number of 2000 iterations, which is similar to the number of evaluations reported in previous studies (e.g., Rakovec *et al.*, 2016, Shen *et al.*, 2022), and only one optimization trial. We have clarified these points in the text of subsection 3.2 (L186-L190).

*Lines 203–205: Clarify why SSI was excluded from specific analyses and define "longer time scales."*

We have removed the expression "longer time scales" and modified the aforementioned sentence as follows to avoid confusion among readers (L214-L217):

"To evaluate how the subjective choice of time scales may affect the characterization of different types of droughts and inter-relationships, we compute SDI-n with n = 1, 3, 6, 9, 12, 18, and 24 months (Figure 2c) excepting the SSI, for which we consider time scales that have been commonly adopted under different assumptions and considerations  (e.g., Núñez et al.,

2014; Oertel et al., 2020; Tijdeman et al., 2020; Baez-Villanueva et al., 2024; see section 3.4).".

*Figures:*
*Figure 1: Add yellow catchment indicators to the legend. Standardize the y-axis scale for mm/month and temperature to enable better comparison between catchments.*

We appreciate the reviewer's suggestion. A new version of Fig. 1 is displayed below, using the same range for the primary y-axis in all the catchments, as suggested by this reviewer. Since monthly precipitation and streamflow values are very difficult to distinguish for the Cochiguaz and Choapa River basins, we prefer to keep the original version of Fig. 1, as it provides a better visualization of the annual cycles in each basin. Additionally, we have modified the caption of Fig.1 in the main document as follows:

"Location, delimitation (orange area in map) and seasonality of precipitation (P), runoff (Q) and temperature for the six case study basins: (a) Cochiguaz River at El Peñón, (b) Choapa River at Cuncumén, (c) Claro River at El Valle, (d) Palos River at Colorado, (e) Ñuble River at La Punilla, and (f) Cautín River at Rari-Ruca. Overlines represent annual averages for the period April/1985-March/2015."

[Figure]

**Figure 1: Same as in Fig. 1 in the main document, but with the same range for the y-axis, as suggested by the reviewer.**

*Figure 2: Define all acronyms directly in the figure legend (e.g., pp, temp, Kin, SPI, SPEI, etc.). Avoid phrases like "See text for details." The figure caption should be self-contained.*

In response to this suggestion, we have revised all figure captions, removing the phrases "see text for details", and defining all the acronyms in the figure captions, as suggested by this reviewer. In particular, we have modified the caption of Figure 2 as follows:

"Figure 2. Flowchart describing the approach used in this study, including: (a) meteorological forcings, hydrological model structure and river routing configuration (section 3.1); (b) calibration and evaluation of hydrological models (section 3.2); (c) calculation of drought indices at different time scales (section 3.3); (d) correlation analysis between standardized drought indices and aggregated fluxes/storages (section 3.4); and (e) drought propagation analysis (section 3.5). The abbreviations/acronyms used in the figure are as follows: P – precipitation; T – air temperature; Kin – incoming shortwave radiation; ap – atmospheric pressure; rh – relative humidity; w – wind speed; SPI – Standardized Precipitation Index; SPEI – Standardized Precipitation and Evapotranspiration Index; SSMI – Standardized Soil Moisture Index; SWSI – Standardized Water Storage Index; SSI – Standardized Streamflow Index."

Also, a new version of Fig. 2 is displayed below, in which the acronym "pp" has been changed to "P" when referring to precipitation, in order to maintain clarity and consistency with the rest of the document.

[Figure]

**References**

Adeyeri, O. E., Zhou, W., Laux, P., Ndehedehe, C. E., Wang, X., Usman, M. &

Akinsanola, A. A. (2023) Multivariate Drought Monitoring, Propagation, and Projection Using Bias-Corrected General Circulation Models. *Earth's Futur.* **11**(4), 1–16. doi:10.1029/2022EF003303

Bachmair, S., Stahl, K., Collins, K., Hannaford, J., Acreman, M., Svoboda, M., Knutson, C., et al. (2016) Drought indicators revisited: the need for a wider consideration of environment and society. *Wiley Interdiscip. Rev. Water* **3**(4), 516–536. doi:10.1002/wat2.1154

Baez-Villanueva, O. M., Zambrano-Bigiarini, M., Miralles, D. G., Beck, H. E., Siegmund, J. F., Alvarez-Garreton, C., Verbist, K., et al. (2024) On the timescale of drought indices for monitoring streamflow drought considering catchment hydrological regimes. *Hydrol. Earth Syst. Sci.* **28**(6), 1415–1439. doi:10.5194/hess-28-1415-2024

Barker, L. J., Hannaford, J., Chiverton, A. & Svensson, C. (2016) From meteorological to hydrological drought using standardised indicators. *Hydrol. Earth Syst. Sci.* **20**(6), 2483–2505. doi:10.5194/hess-20-2483-2016

Bevacqua, A. G., Chaffe, P. L. B., Chagas, V. B. P. & AghaKouchak, A. (2021) Spatial and temporal patterns of propagation from meteorological to hydrological droughts in Brazil. *J. Hydrol.* **603**(PA), 126902. Elsevier B.V. doi:10.1016/j.jhydrol.2021.126902

Bhardwaj, K., Shah, D., Aadhar, S. & Mishra, V. (2020) Propagation of Meteorological to Hydrological Droughts in India. *J. Geophys. Res. Atmos.* **125**(22). doi:10.1029/2020JD033455

Carrão, H., Russo, S., Sepulcre, G. & Barbosa, P. (2013) Agricultural Drought Assessment In Latin America Based On A Standardized Soil Moisture Index. *ESA Living Planet Symp.* (December).

Clark, M. P., Nijssen, B., Lundquist, J. D., Kavetski, D., Rupp, D. E., Woods, R. A., Freer, J. E., Gutmann, E. D., Wood, A. W., Brekke, L. D., et al. (2015) A unified approach for process-based hydrologic modeling: 1. Modeling concept. *Water Resour. Res.* doi:10.1002/2015WR017198

Clark, M. P., Nijssen, B., Lundquist, J. D., Kavetski, D., Rupp, D. E., Woods, R. A., Freer, J. E., Gutmann, E. D., Wood, A. W., Gochis, D. J., et al. (2015) A unified approach for process-based hydrologic modeling: 2. Model implementation and case studies. *Water Resour. Res.* doi:10.1002/2015WR017200

Clark, M. P., Zolfaghari, R., Green, K. R., Trim, S., Knoben, W. J. M., Bennett, A., Nijssen, B., et al. (2021) *The numerical implementation of land models: Problem formulation and laugh tests. J. Hydrometeorol.*, Vol. 22, 1627–1648. doi:10.1175/JHM-D-20-0175.1

Fowé, T., Yonaba, R., Mounirou, L. A., Ouédraogo, E., Ibrahim, B., Niang, D., Karambiri, H., et al. (2023) From meteorological to hydrological drought: a case study using standardized indices in the Nakanbe River Basin, Burkina Faso. *Nat. Hazards* (0123456789). Springer Netherlands. doi:10.1007/s11069-023-06194-5

Garcia, F., Folton, N. & Oudin, L. (2017) Which objective function to calibrate rainfall–runoff models for low-flow index simulations? *Hydrol. Sci. J.* **62**(7), 1149–1166. Taylor & Francis. doi:10.1080/02626667.2017.1308511

Gautam, S., Samantaray, A., Babbar-Sebens, M. & Ramadas, M. (2024) Characterization and Propagation of Historical and Projected Droughts in the Umatilla River Basin, Oregon, USA. *Adv. Atmos. Sci.* **41**(2), 247–262. doi:10.1007/s00376-023-2302-8

Gobron, N., Pinty, B., Mélin, F., Taberner, M., Verstraete, M. M., Belward, A., Lavergne, T., et al. (2005) The state of vegetation in Europe following the 2003 drought. *Int. J.*

*Remote Sens.* **26**(9), 2013–2020. doi:10.1080/01431160412331330293

Hoffmann, D., Gallant, A. J. E. & Arblaster, J. M. (2020) Uncertainties in Drought From Index and Data Selection. *J. Geophys. Res. Atmos.* **125**(18), 1–21. doi:10.1029/2019JD031946

Huang, S., Li, P., Huang, Q., Leng, G., Hou, B. & Ma, L. (2017) The propagation from meteorological to hydrological drought and its potential influence factors. *J. Hydrol.* **547**, 184–195. Elsevier B.V. doi:10.1016/j.jhydrol.2017.01.041

Loon, A. Van & Laaha, G. (2015) Hydrological drought severity explained by climate and catchment characteristics. *J. Hydrol.* **526**, 3–14. Elsevier B.V. doi:10.1016/j.jhydrol.2014.10.059

McKee, T. B., Doesken, N. J. & John, K. (1993) The relationship of drought frequency and duration to time scales. *Eighth Conf. Appl. Climatol.* Anaheim, California.

Mizukami, N., Clark, M. P., Gharari, S., Kluzek, E., Pan, M., Lin, P., Beck, H. E., et al. (2021) A Vector-Based River Routing Model for Earth System Models: Parallelization and Global Applications. *J. Adv. Model. Earth Syst.* **13**(6), 1–20. doi:10.1029/2020MS002434

Mizukami, N., Clark, M. P., Sampson, K., Nijssen, B., Mao, Y., McMillan, H., Viger, R. J., et al. (2016) mizuRoute version 1: a river network routing tool for a continental domain water resources applications. *Geosci. Model Dev.* **9**(6), 2223–2238. doi:10.5194/gmd-9-2223-2016

Núñez, J., Rivera, D., Oyarzún, R. & Arumí, J. L. (2014) On the use of Standardized Drought Indices under decadal climate variability: Critical assessment and drought policy implications. *J. Hydrol.* **517**, 458–470. Elsevier B.V. doi:10.1016/j.jhydrol.2014.05.038

Odongo, R. A., Moel, H. De & Loon, A. F. Van. (2023) Propagation from meteorological to hydrological drought in the Horn of Africa using both standardized and threshold-based indices. *Nat. Hazards Earth Syst. Sci.* **23**(6), 2365–2386. doi:10.5194/nhess-23-2365-2023

Oertel, M., Meza, F. J. & Gironás, J. (2020) Observed trends and relationships between ENSO and standardized hydrometeorological drought indices in central Chile. *Hydrol. Process.* **34**(2), 159–174. doi:10.1002/hyp.13596

Palmer, W. C. C. (1965) Meteorological Drought. *U.S. Weather Bur. Res. Pap. No. 45* **30**, 58.

Peña-Gallardo, M., Vicente-Serrano, S. M., Hannaford, J., Lorenzo-Lacruz, J., Svoboda, M., Domínguez-Castro, F., Maneta, M., et al. (2019) Complex influences of meteorological drought time-scales on hydrological droughts in natural basins of the contiguous Unites States. *J. Hydrol.* **568**(November 2018), 611–625. Elsevier. doi:10.1016/j.jhydrol.2018.11.026

Rakovec, O., Kumar, R., Attinger, S. & Samaniego, L. (2016) Improving the realism of hydrologic model functioning through multivariate parameter estimation. *Water Resour. Res.* **52**(10), 7779–7792. doi:10.1002/2016WR019430

Rivera, J. A., Otta, S., Lauro, C. & Zazulie, N. (2021) A Decade of Hydrological Drought in Central-Western Argentina. *Front. Water* **3**(April), 1–20. doi:10.3389/frwa.2021.640544

Roo, A. P. J. De, Wesseling, C. G. & Deursen, W. P. A. Van. (2000) Physically based river basin modelling within a GIS: the LISFLOOD model. *Hydrol. Process.* **14**(11–12), 1981–1992. doi:10.1002/1099-1085(20000815/30)14:11/12<1981::AID-

HYP49>3.0.CO;2-F

Seibert, M., Merz, B. & Apel, H. (2017) Seasonal forecasting of hydrological drought in the Limpopo Basin: a comparison of statistical methods. *Hydrol. Earth Syst. Sci.* **21**(3), 1611–1629. doi:10.5194/hess-21-1611-2017

Sepulcre-Canto, G., Horion, S., Singleton, A., Carrao, H. & Vogt, J. (2012) Development of a Combined Drought Indicator to detect agricultural drought in Europe. *Nat. Hazards Earth Syst. Sci.* **12**(11), 3519–3531. doi:10.5194/nhess-12-3519-2012

Shen, H., Tolson, B. A. & Mai, J. (2022) Time to Update the Split-Sample Approach in Hydrological Model Calibration. *Water Resour. Res.* **58**(3), 1–26. doi:10.1029/2021wr031523

Stahl, K., Vidal, J. P., Hannaford, J., Tijdeman, E., Laaha, G., Gauster, T. & Tallaksen, L. M. (2020) The challenges of hydrological drought definition, quantification and communication: An interdisciplinary perspective. *Proc. Int. Assoc. Hydrol. Sci.* **383**, 291–295. doi:10.5194/piahs-383-291-2020

Svoboda, M & Buchs, B. (2016) Handbook of drought indicators and indices.

Svoboda, Mark, LeComte, D., Hayes, M., Heim, R., Gleason, K., Angel, J., Rippey, B., et al. (2002) THE DROUGHT MONITOR. *Bull. Am. Meteorol. Soc.* **83**(8), 1181–1190. doi:10.1175/1520-0477-83.8.1181

Teutschbein, C., Quesada Montano, B., Todorović, A. & Grabs, T. (2022) Streamflow droughts in Sweden: Spatiotemporal patterns emerging from six decades of observations. *J. Hydrol. Reg. Stud.* **42**(June). doi:10.1016/j.ejrh.2022.101171

Tijdeman, E., Stahl, K. & Tallaksen, L. M. (2020) Drought Characteristics Derived Based on the Standardized Streamflow Index: A Large Sample Comparison for Parametric and Nonparametric Methods. *Water Resour. Res.* **56**(10). doi:10.1029/2019WR026315

Vicente-Serrano, S. M., Beguería, S. & López-Moreno, J. I. (2010) A multiscalar drought index sensitive to global warming: The standardized precipitation evapotranspiration index. *J. Clim.* **23**(7), 1696–1718. doi:10.1175/2009JCLI2909.1

Wan, W., Zhao, J., Li, H. Y., Mishra, A., Hejazi, M., Lu, H., Demissie, Y., et al. (2018) A Holistic View of Water Management Impacts on Future Droughts: A Global Multimodel Analysis. *J. Geophys. Res. Atmos.* **123**(11), 5947–5972. doi:10.1029/2017JD027825

Wang, F., Wang, Z., Yang, H., Di, D., Zhao, Y., Liang, Q. & Hussain, Z. (2020) Comprehensive evaluation of hydrological drought and its relationships with meteorological drought in the Yellow River basin, China. *J. Hydrol.* **584**(June 2019), 124751. Elsevier. doi:10.1016/j.jhydrol.2020.124751

Wu, J., Chen, X., Yao, H., Gao, L., Chen, Y. & Liu, M. (2017) Non-linear relationship of hydrological drought responding to meteorological drought and impact of a large reservoir. *J. Hydrol.* **551**, 495–507. Elsevier B.V. doi:10.1016/j.jhydrol.2017.06.029

Wu, J., Yao, H., Chen, X., Wang, G., Bai, X. & Zhang, D. (2022) A framework for assessing compound drought events from a drought propagation perspective. *J. Hydrol.* **604**(November 2021), 127228. Elsevier B.V. doi:10.1016/j.jhydrol.2021.127228

Yun, X., Tang, Q., Wang, J., Li, J., Li, Y. & Bao, H. (2023) Reservoir operation affects propagation from meteorological to hydrological extremes in the Lancang-Mekong River Basin. *Sci. Total Environ.* **896**(July), 165297. Elsevier B.V. doi:10.1016/j.scitotenv.2023.165297

Zhang, X., Hao, Z., Singh, V. P., Zhang, Y., Feng, S., Xu, Y. & Hao, F. (2022) Drought

propagation under global warming: Characteristics, approaches, processes, and controlling factors. *Sci. Total Environ.* **838**(19), 156021. Elsevier B.V. doi:10.1016/j.scitotenv.2022.156021